# Role of New Chiral Additives on Physical-Chemical Properties of the Nematic Liquid Crystal Matrix

**DOI:** 10.3390/ma16176038

**Published:** 2023-09-02

**Authors:** Alexey S. Merekalov, Oleg N. Karpov, Georgiy A. Shandryuk, Olga A. Otmakhova, Alexander V. Finko, Artem V. Bakirov, Vladimir S. Bezborodov, Raisa V. Talroze

**Affiliations:** 1A.V. Topchiev Institute of Petrochemical Synthesis of the Russian Academy of Sciences, 29 Leninsky P., Moscow 119991, Russia; 2Department of Chemistry, Moscow State University, Leninskie Gory 1/3, Moscow 119991, Russia; 3N.S. Enikolopov Institute of Synthetic Polymeric Materials, Russian Academy of Sciences, 70 Profsoyuznaya Str., Moscow 117393, Russia; 4Department of Chemistry, Belarusian State Technological University, 13a Sverdlova Street, 220006 Minsk, Belarus

**Keywords:** liquid crystal, optical activity, chirality, additive, dielectric properties

## Abstract

We have synthesized and studied three new chiral substances as additives to a nematic liquid crystal. The difference in the optical activity and chemical structure of additive molecules results in the appearance of the chiral nematic phase and the change in both the compatibility of the mixture components and temperature range of the liquid crystal phase. The role of additives with fundamentally different structures and optical activities is shown. The increase in the *T*_NI_ that is observed in mixtures with 4-[(2*S*)-(+)-2-Methylbutoxy]benzoic acid indicate the possibility of the increase in order caused by the formation of molecularly rigid and elongated dimers of the additive, which was confirmed using infrared spectra. The doping of the nematic liquid crystal with (2*R*)-(+)-2-[4-[2-Chloro-4-(4-hexylphenyl)phenyl]phenoxy]propanoic acid causes the lowering of *T*_NI._ The binol derivative *S-*(+)-6-[1-[2-(5-Carboxypentoxy)naphthalen-1-yl]naphthalen-2-yl] oxyhexanoic acid has the highest chirality among the additives used. One can explain the effects observed in terms of the role of size, shape, and compatibility with the nematic matrix as shown by the molecules that are used as additives.

## 1. Introduction

For many years, liquid crystal (LC) systems have been of great interest due to their unique properties and possible applications. This is particularly true for chiral liquid crystals [1]. The current and possible applications of such LC systems include photonics [2], electro-optics [3], chromatographic stationary phases with high chiral selectivity [4], chemical sensors [5], etc. Chiral LCs may be created due to the introduction of the chiral asymmetrical center in mesogenic molecules or through the insertion of mesogenic or non-mesogenic chiral dopant in a liquid crystal phase. The latter method indicates that a chiral dopant transfers a molecular chirality on the phase organization. It is evident that the relationship between the molecular structure of a liquid crystal and dopant and the intermolecular interactions must play an important role. The question arises as to how a molecular chirality may be transferred into the chirality of a bulky LC. Several groups prove and discuss this problem [1,6,7,8,9,10,11,12,13], although there is no strict theoretical solution as of yet; however, there are several model attempts published in [7,14,15,16,17]. The basic approach considers the interplay between the molecular shape of the chiral dopant and the elastic properties of the nematic LC phase, which controls the restoring torques opposing the distortion of the director.

The major goal of the current research work is to study the effect of three newly synthesized chiral additives (CAs), which have different molecular shapes and optical activities on the optical, thermal, and dielectric properties of a nematic (N) liquid crystal when mixed in. The optical activity is the property of a compound being able to rotate the plane of polarization of plane-polarized light, and a compound with such activity is labelled as optically active. We have chosen a new set of chiral substances, namely, 4-[(2*S*)-(+)-2-Methylbutoxy]benzoic acid (**CA1**), (2 *R*)-(+)-2-[4-[2-Chloro-4-(4-hexylphenyl)phenyl]phenoxy]propanoic acid (**CA2**), and *S*-(+)-6-[1-[2-(5-Carboxypentoxy)naphthalen-1-yl]naphthalen-2-yl]oxyhexanoic acid (**CA3**) (Figure 1).

We show in this paper that these systems are optically active, and we intend to figure out their ability to induce the chiral structure of the low molecular nematic matrix (LMN), which is the mixture of several derivatives of 4-cyanobiphenyl. This matrix has a broad nematic temperature range, and it is widely used in our research. We have presented the whole description of LMN composition in [18]. The major difference in the chiral additives’ molecular structures dictates the necessity for the analysis of both the principal compatibility of components on the phase behavior and the other physical-chemical properties of the resultant mixtures. One expects that at first, it will affect the optical properties of new LC mixtures. Going forward, we are considering these systems to be effective if the additives used as materials for electro-optic displays provide a strong twisting of the nematic matrix and preserve the necessary orientation control in the cell. Moreover, we consider new additives as possible ligands, transfer the chirality to quantum dots, ensure their compatibility with nematic liquid crystals, and provide their reasonable effect on the switching times of optoelectronic effects.

## 2. Materials and Methods

### 2.1. Synthesis of Chiral Additives

#### 2.1.1. 4-[(2S)-(+)-2-Methylbutoxy]benzoic Acid (**CA1**)

**CA1** is prepared through the alkylation of ethyl ester 4-hydroxybenzoic acid and the alkaline hydrolysis of the reaction product. The yield is 77%. Figure 1 of the synthetic route is given below.

#### 2.1.2. (2R)-(+)-2-[4-[2-Chloro-4-(4-hexylphenyl)phenyl]phenoxy]propanoic Acid (**CA2**)

**CA2** was prepared from the corresponding 3,6-disubstituted cyclohex-2-enone (**1**). Along the route given in Figure 2. An amount of 0.1 mol of 4”Hexyl-2′-chloro-4-hydroxyterphenyl (**3**), 0.11 mol of ethyl ester of S-lactic acid, and 0.12 mol of triphenylphosphine were dissolved in 200 mL of dry THF and cooled to 0 °C. Then, 0.12 mol of diethyl azodicarboxylate was added in small portions at 0 °C. The mixture was allowed to stir overnight at room temperature. THF was then distilled out in vacuo, and the product (**4**) was purified using column chromatography on silica gel (eluent petroleum ether-ethyl acetate: 10:1) with a yield of 69%.

We dissolved 0.05 mol of ethyl ester of (**4**) in a mixture of 30 mL of THF, 30 mL of ethyl alcohol, and 5 mL of water. Then, 0.11 mol of LiOH was added and stirred at room temperature for 12 h. The reaction mixture was diluted with water and acidified to pH = 6. The product **CA2** was extracted using dichloromethane. The extract was dried using anhydrous sodium sulfate. Dichloromethane was evaporated. The expected product **CA2** was obtained with a yield of 85%.

#### 2.1.3. S-(+)-6-[1-[2-(5-Carboxypentoxy)naphthalen-1-yl]naphthalen-2-yl]oxyhexanoic Acid (**CA3**)

There were two steps in the synthetic route: (a). Diethyl ester of 6,6′-([1,1′-Binaphthalene]-2,2′-diylbis(oxy))dihexanoic acid was synthesized from 1,1′-Bi-2-naphthol (0.01 mol), which was mixed with ethyl 6-bromohexanoate (0.023 mol), 5.5 g of K_2_CO_3_, and 0.16 g of KI in 50 mL of methylethylketone. The mixture was refluxed for 22 h, cooled down to room temperature, and poured into water. We have twice extracted the product with methylene chloride. The combined organic layers were washed with water and dried over magnesium sulfate. After removing the solvent, we purified the product using column chromatography on silica gel (eluent petroleum ether-ethyl acetate) with a yield of 87%; and (b). 6,6′-([1,1′-Binaphthalene]-2,2′-diylbis(oxy))dihexanoic acid was obtained from the diethyl ester of 6,6′-([1,1′-Binaphthalene]-2,2′-diylbis(oxy))dihexanoic acid (0.005 mol) dissolved in a mixture of 30 mL of THF, 30 mL of ethyl alcohol, and 5 mL of water. Then, 0.015 mol of LiOH was added, and the mixture was stirred at room temperature for 12 h. The reaction mixture was diluted with water and acidified to pH = 6. The final product was extracted with dichloromethane and dried with anhydrous sodium sulfate. After removing the solvent, the product was also purified using column chromatography on silica gel (eluent petroleum ether-ethyl acetate: 7:1) with a yield of 77%.

### 2.2. Methods Used

Bruker Avance 400 (Bruker Scientific Instruments, Billerica, MA, USA) and Agilent 400-MR instruments (Agilent Technologies, Inc., Santa Clara, CA, USA) allowed us to measure nuclear magnetic resonance (NMR) spectra at room temperature.

High-resolution electrospray ionization mass spectrometry was performed using a Bruker microTOF II instrument.

The elemental analysis of CHNS/O was performed using a Perkin Elmer Model 2400 Series II (Perkin Elmer, Waltham, MA, USA).

A CM-3 circular polarimeter (ZOMZ, Sergiev Posad, Russia) was used to measure the angle of rotation of the polarization plane of solutions of optically active additives in THF at a wavelength of 589 nm in the range of ± 90 degrees with an accuracy of ± 0.04 degrees.

Differential scanning calorimetry (DSC) measurements were carried out using a Mettler DSC 823E (Mettler Toledo, Greifensee, Switzerland) instrument using 40 μL Al pans at a heating rate of 10 °C/min in an argon atmosphere. The melting points and enthalpies of indium and zinc were used for temperature and heat capacity calibration.

FTIR spectra were recorded using IFS 66 v/s spectrometer (“Bruker”) (30 scans, resolution 1 cm^−1^, wavelength range 400–4000 cm^−1^). Samples were prepared in the form of thin layers located between two KBr optical glasses.

Polarization optical microscopy (POM) micrographs were taken on a Polam L-213 microscope (LOMO, St. Petersburg, Russia) equipped with a Mettler Toledo FP-82 HT hot stage and a Mettler Toledo FP-90 temperature control unit. The temperature of the tested samples was maintained at an accuracy of 0.1 °C. Samples were placed on microscope slides (Fisherbrand™, Fisher Scientific, Pittsburgh, PA, USA) and covered with cover slips. The surfaces of the slides were not specifically treated. The heating/cooling rate was 5 °C/min.

Selective reflection spectra at different temperatures were obtained on the aforementioned Polam L-213 microscope through an additional output to an optical fiber connected to a USB-2000 spectrometer (Ocean Optics, Largo, FL, USA). The spectra were processed using SpectraSuite 1.0 software (Ocean Optics, USA) in the “Transmission Measurement” mode.

For dielectric measurements, the temperature measurements of the dielectric characteristics of the system under study were carried out in a self-made cell in the form of a flat capacitor with a gap of 5 μm, which was constructed using glass plates coated with indium tin oxide (ITO) with a surface resistance of less than 15 Ω. For measuring equipment, we used an LCR Meter ET4510 (at fixed frequency 10 kHz 100 mV signal level) (Hangzhou Zhongchuang Electron Co., Ltd., Hangzhou, China) in an isothermal regime. We provided a planar orientation of LC samples by mechanically rubbing the polyimide films covering the cell surface.

Small-angle X-ray high-resolution diffraction patterns were recorded using an S3-Micropix SAXS camera (Hecus, Graz, Austria). A two-dimensional Dectris Pilatus 100K detector was employed for data collection; the high voltage and current at Xenocs Genix generator were set to 50 kV and 1 mA (CuK**_α_**, λ = 1.542 Å), respectively. For the shaping of the X-ray beam, Fox 3D vacuum optics were used, and the slits in the Kratky collimator were set to 0.1 and 0.2 mm, respectively. To calibrate small-angle diffractograms, silver behenate calibrant was used as a reference. To eliminate the influence of air, the X-ray optics system and camera was vacuumed to a pressure of 0.89 Pa. The exposition time was 3000 s. Temperature variation was carried out by a Peltier hot stage with a temperature precision of 0.11 °C.

The samples were prepared by dissolving the required amount of additive (**CA1-CA3**) in the LC matrix (LMN) and mixing at a temperature above *T*_NI_, until a completely homogeneous mixture was obtained.

## 3. Results and Discussion

The Materials and Methods section contains a detailed description of the synthesis of new chiral additives. The quality of synthesized substances was checked with IR spectra (Appendix A) and different NMR and mass spectral analyses. Several examples are provided in Appendix A. The results of the CHNS/O elemental analysis are presented in Appendix A. The first two dopants out of three are crystalline. This is proven by the crystalline textures of **CA1** and **CA2** (Figure 2a), which undergo a transition to another crystal with a different texture. The dopants are melting to form isotropic liquids with the heat of fusion values given in Table 1. The DSC curves of all three additives are given in Figure 2b. The crystalline state of **CA1** and **CA2** is also confirmed by endothermic peaks (curves 1,2) that accompany the melting under heating. As for the dopant **CA3**, it forms a glassy state and turns into a liquid at glass transition temperature *T*_g_, i.e., 17 °C (Figure 2b, curve 3). Table 1 summarizes the corresponding transition temperatures and melting enthalpies.

These compounds are optically active and, as shown in Table 1, have different values for their specific optical rotation, namely ranging from +12 to +43 deg mL g^−1^ dm^−1^. The first two compounds contain an asymmetrical carbon atom that is responsible for the optical activity, and the other compound is the binol derivative **CA3**, which has axial optical asymmetry that contributes to the optical activity (atropisomerism).

Appendix A presents the IR spectra of additives, which reflect the structure of the additives. One can mention the p-substituted aromatic rings and carboxylic groups (marked in the spectra) and ether Ph-O-CH_2_- groups that are presented as intensive peaks in 1250–1000 cm^−1^ range. Intensive peaks at 2870–2960 cm^−1^ and 1460 cm^−1^ are characteristic for polymethylene chains in the **CA2** and **CA3** molecules, and the band splitting related to phenyl rings in the IR spectrum of **CA3** confirms the presence of naphthalene rings. A particularly important point is that the IR spectra show that **CA1** forms dimers due to hydrogen bonds between carboxylic groups (C=O at 1680 cm^−1^), whereas **CA2** prefers the monomeric structure.

The introduction of optically active additives into liquid crystals assumes the induction of chiral phases, primarily chiral nematics (N*). In this case, we must find out a sufficiently good compatibility of components and elucidate the effect of the additive content on the transition temperature from the nematic to isotropic (N–I) phase. As an example, Appendix A demonstrates the DSC curves of LMN composites with **CA1**, **CA2**, and **CA3**. Figure 3 presents the change in *T*_NI_ with the increase in the CA component content for all systems under study. The comparison *T*_NI_ as a function of the mixture content (Figure 3a) shows a continuous increase in the *T*_NI_ (curve 1) and its decrease (curve 2). One can see that the N–I phase transition temperature may either increase or decrease with the addition of **CA1** and **CA2**. The increase in *T*_NI_ observed in mixtures with **CA1** indicate the possibility of the increase in order that is caused by the formation of molecularly rigid and elongated dimers of **CA1** in the nematic LC, as confirmed by the infrared spectra. **CA1** contributes to the increase of the transition enthalpy (from 3.4 up to 6.5 J g^−1^) (Figure 3b, curve 1). At the same time, one may see in LMN mixtures with **CA2** that there is a continued drop in *T*_NI_ without any visible destruction of the LC structure; however, the corresponding enthalpy decreases from 3.4 down to 2.2 J g^−1^ (Figure 3b, curve 2). It is a clear indication that the **CA2** additive destroys LMN, which is possibly due to the large size of its molecules in comparison with the nematic matrix molecules. These data are in a good agreement with the quantitative correlation established in [19]. They show the doping of the liquid crystal with molecularly flexible acids to cause the lowering of *T*_NI._ The increase in *T*_NI_ occurs if the rigid carboxylic acids form dimers are comparable in size with the nematic matrix molecules.

LMN mixtures with **CA3** behave in a quite different way. As shown in Figure 3a, (curve 3) the *T*_NI_ of mixtures drops down to 5 wt.% of an additive and then does not change with the increase in **CA3** content. The enthalpy of the N–I transitions (Figure 3b) changes in a similar way with the increase in **CA3** content. This result makes one consider the phase separation that is occurring in the system with greater than 5 wt.% of **CA3**.

The analysis of the optical textures obtained in the crossed polarizers in cells without surface treatment shows that the layer of LMN has a homeotropic texture, with the long molecular axis being oriented preferably normal to the sample plane (Figure 4a). The addition of **CA1** in amounts creating mixtures with a concentration below 10 wt.% does not change the homeotropic texture, even if the clearing temperature *T*_NI_ is higher (Figure 3a, curve 1). However, the cooling of that sample to below the *T*_NI_ of the mixture (70 °C) results in the appearance of an oil stripe texture. (Figure 4b). At a **CA1** content of below 25 wt.%, the solubility of the initially crystalline **CA1** is sufficiently high and is preserved at room temperature, indicating the weak influence of the chiral dopant on the nematic matrix and the preservation of the homeotropic texture. Roughly speaking, the anchoring energy of the nematic is higher than the twisting power of **CA1**. Nevertheless, all samples with a **CA1** content of below 25 wt.% undergo a violation of the homeotropic orientation during the heating up to the vicinity of *T*_NI_ of LMN matrix. The higher the content of **CA1**, the lower the temperature for the texture change. As for the mixture with 25 wt.% of **CA1**, the homeotropic texture at room temperature is distorted and small light structures appear (Appendix A), which are growing into the chiral texture (Figure 4e). Generally, at **CA1** concentrations ranging between 15 and 25 wt.%, cooling below the *T*_NI_ of the mixtures allows one to observe the polydomain textures being formed by the chiral nematic structures and the chiral twisting being stronger than that of when it is under heating (Appendix A).

Contrary to the LMN-**CA1** mixtures, the *T*_NI_ transition of mixtures with **CA2** decreases (Figure 3b, curve 2) with the increase in **CA2** content. However, mixtures starting with 2 wt.% of **CA2** have a chiral twisted texture at room temperature (Figure 5). The presence of **CA2** seems to change the anchoring of the matrix molecules, which transforms the homeotropic texture. This effect, in combination with the higher optical activity of **CA2** chiral molecules (Table 1), allows for the selective reflection to appear in the visible spectral range.

As an example, Figure 6a shows the curves related to the selective reflection of light of the LMN with 30 wt.% of **CA2** [18]. When the content of the additive reaches 20 wt.%, the selective reflection happens at 850–950 nm (Figure 6b). At 30 wt.% **CA2**, the selective reflection is observed at 500–600 nm, whereas at 40 wt.%, it shifts down to 400–500 nm.

At the same time, the optical activity of **CA3** is already two times higher than that of **CA2** (Table 1); that is why it induces chiral structure of the mixture at 0.5 wt.% at room temperature already (Figure 7a). The chiral texture exists for all of the mixture contents studied (Figure 7), but at 6 wt.% **CA3** and higher, one may observe the appearance of dark areas, which indicate the secretion of the excess of amorphous areas of **CA3** liquid. This means that the compatibility of the components becomes poor, and dark areas co-exist with the chiral texture. The maximum saturation of **CA3** by a nematic matrix seems to be about 5 wt.%. The concentration range where **CA3** shows full compatibility with LMN is quite narrow.

The ITO surfaces organize the nematic matrix with the formation of homeotropic orientation (Figure 4a). However, the treatment of the surface with the planar surface orientant results in the formation of LMN planar texture. Figure 8a shows the corresponding curves of the dielectric permittivity as a function of temperature. As for mixtures with **CA3**, a small amount of the additive also induces the planar orientation without any prior treatment of ITO. The dielectric permittivity of the mixtures in the planar texture continuously increases with temperature leading up to *T*_NI_, and then it becomes independent of temperature or slightly decreases. The latter is typical for low molecular liquid crystals in an isotropic phase.

As is observed from the optical textures, mixtures with **CA2** may form an initial planar orientation similar to those with **CA3**. This is confirmed with the curve ε′/ε′_is_(T) of 5 wt.% mixture with **CA2** (Figure 8b). At a higher **CA2** content (about 20 wt.%), the dielectric permittivity behaves in a similar way while being heated. However, during the cooling of the sample from the isotropic phase, the value of the ε′/ε′_is_ begins to increase below *T*_NI_. Notably, the LC optical texture practically does not change when the mixture is heated or cooled (Appendix A). It seems that this effect may be related to the orienting influence of the field signal level on the LC composition with a positive dielectric anisotropy. Along with an increase in **CA2** content, which induces stronger chiral twisting, the anchoring effect becomes much weaker. This results in a partial molecular reorientation along the field direction during the cooling from the isotropic phase.

Contrary from the above the mixtures, the ones containing **CA1** do not reorient, and the homeotropic matrix orientation stays the same for the mixtures. That is why we have analyzed the dielectric constant dependence with temperature with the use of the surface orientant, which makes the initial liquid crystal mixture plane oriented for all concentrations of **CA1**. Heating of the samples up to 10% results in the continuous increase in the relative dielectric permittivity ε′/ε′_is_ (Figure 9).

The conditions of the experiment were as follows: the optical textures and dielectric measurements were simultaneously analyzed in cells with unidirectionally treated surfaces.

The analysis of the optical textures as a function of the dopant content allows us to hypothesize that, at concentrations below 25 wt.%, the planar unidirectional anchoring is preserved on both surfaces. The mixtures exhibit very similar twisted structures with changing temperatures and concentrations. At higher concentrations such as 25 wt.% and above, the set of various optical textures becomes broader. It may be likely that the planar anchoring is a degenerate one.

The initial homogeneous texture of the LMN matrix (Figure 10a) does not really change when one inserts 10 wt.% of **CA1** (Figure 10b). When approaching the *T*_NI_ of this mixture, the chiral texture appears, which exists in a very narrow temperature range (Figure 10c). The increase in **CA1** content up to 20 wt.% also does not change the initial planar orientation (Figure 10d). However, when approaching the *T*_NI_ of the initial nematic matrix (64 °C) by heating, the typical twisted texture appears with the axis located in the in-plane of the cell (Figure 10e). One may suppose that at that moment, the interaction between the surface and mesogenic molecules is becoming weaker. At the same time, a small peak in the ε′/ε′_is_(T) curve (Figure 9) appears. With a further increase in **CA1** content up to 25 wt.%, the peak becomes more pronounced, and a second peak shows up in the vicinity of 72 °C. The corresponding optical textures are shown in Figure 10f–j. The analysis of the textures lets us draw conclusions about the transformation of the N structure of the mixture to a more ordered one (Figure 10f). Above 64 °C, the ordered texture is transformed to a chiral one, which also changes with the temperature (Figure 10g,h). Note that in all cases, this system forms the planar orientation with the characteristic texture known as oil striping (Figure 10h). At 72 °C, the peak on the ε′/ε′_is_(T) curve indicates the next transformation of a chiral texture, which is accompanied by the relocation of the twisting axis into the plane of the cell (Figure 10i,j). The cause of the change in the location of the axis from the normal to a parallel cell surface is not yet clear. One of the possible explanations may be the stronger twisting effect from heating (Figure 10i). When approaching the *T*_NI_ of the 25 wt.% composition, the texture changes again to the planar one with characteristic oil stripes (Figure 10j).

As for the mixture with 30 wt.% of **CA1**, its curve has only one peak (Figure 9) at about 70 °C. This case corresponds to the transformation of the smectic-like fan texture (Figure 10k), which exists in a broad temperature range, into a different texture, which resembles the blue phase texture (Figure 10l). The latter endures the transition to planar textures with oil stripes, preceding the transition to an isotropic phase at 85 °C (Figure 10m).

The analysis of the optical textures as a function of the dopant content allows us to suppose that at concentrations below 25 wt.%, the planar unidirectional anchoring is preserved on both surfaces, and the mixtures exhibit very similar twisted structures when undergoing a change in temperature and concentration. At higher concentrations such as 25 wt.% and above, the set of various optical textures becomes broader. It may be likely that the planar anchoring is a degenerate one because of the increase in **CA1** dopant content.

The texture transitions discussed above are in good agreement with the DSC data (Figure 11). There are two transitions observed: during heating, the first transition corresponds to 72 °C with a transition enthalpy of 4 J g^−1^, and the second one proceeds at 84.6 °C with a corresponding enthalpy of 5.5 J g^−1^ (curve 1). These transitions are fully reversible, as shown in Figure 11, curve 2.

Thus, by using the POM and DSC results, we observe the transition of the 30 wt.% system with a fan texture to the chiral phase, which in its turn is transformed in the isotropic phase. To analyze the transition observed, we have used a small-angle X-ray scattering analysis. The curves obtained at various temperatures are given in Figure 12. One can observe an X-ray Bragg reflection with the corresponding d-spacing equal to 2.31 nm at room temperature. This d-spacing is roughly the size of paired **CA1** molecules, which are connected by a hydrogen bond. It is a supporting evidence of the presence of a layered mesophase.

The increase in temperature up to 50 °C leads to the temperature shift of the small-angle maximum to a d-spacing of 2.33 nm, which corresponds to the linear thermal expansion coefficient of β = 3.6 × 10^−4^ K^−1^ in the material, typical for LC. However, further heating leads to the drop in the diffraction intensity, and the maximum completely disappears at 72 °C. This leads one to conclude that the increase in **CA1** content leads the nematic phase at lower contents to be transformed in to a smectic one. While no second or third order reflections were detected, this pattern still can be attributed to the smectic ordering due to the narrow peak and the type of the texture (see Figure 10k). Thus, the first transition is S-N*, and the second one—N*-I.

## 4. Conclusions

The analysis of the structure, thermal, dielectric, and optical properties of materials, which compose the nematic matrix and new chiral additives, allows for the estimation of the capabilities of new chiral dopants to induce optically chiral structure of the nematic liquid crystal phase influencing the phase transitions. Chiral dopants can exhibit a different ability to twist the nematic phase. As mentioned above, there is no strict theoretical solution as of yet that can provide the full explanation of that phenomenon. However, there are some general rules [7], which we have used to describe our choice of chiral additives. The choice of the new additives with good optical activity but very different molecular structures illustrates the influence of the shape, size, and compatibility of the additive molecule on the properties of the resulting material, including the change in the LC phase range. The additive molecule with a small size (**CA1**) provides the maximal compatibility with the LC matrix and demonstrates the increase in the temperature of the nematic–isotropic transition. Among a wide range of known chiral dopants that induce twisting of the nematic phase of a low molecular weight LC, the latter is very unusual and extremely rare. In most cases, if the size of the additive molecules is bigger than that of molecules of the matrix, the N–I temperature falls down with the increase in the additive content. This result corresponds to the other **CA2** additive. However, its higher optical activity results in the selective reflection of light in the visible wavelength range, which indicates a stronger twisting of the cholesteric pitch. The high optical activity of the third additive (**CA3**) allows for helical twisting at a very low content level, but the difference in the molecular structure provides strong limitations on the compatibility. Unfortunately, there is no solid theoretical model that takes into account different factors of chiral additives to predict the quantitative regularities of the twisting process. That is why it is important to consider the role of additives with fundamentally different structures and optical activities.

## Data Availability

Not applicable.

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
