# Peer review of "Role of New Chiral Additives on Physical-Chemical Properties of the Nematic Liquid Crystal Matrix"

_materials, 2023, doi:10.3390/ma16176038_

Round 1
Reviewer 1 Report
1. The authors have studied three new chiral substances as additives to a nematic liquid crystal. The difference in the optical activity and the chemical structure of additive molecules results in the appearance of the chiral nematic phase, in the change in both compatibility of mixture components and the temperature range of a liquid crystal phase. The paper is well-written, in general. The findings were supported by extensive measurements with a variety of instrumentation.
2. A survey of chiral additives that have been studied should be added. What is the logic of leading to the selection of these additives?
3. The authors showed that the additives resulted in nematic mixtures with somewhat different characteristics. What have we learned from these studies? The authors should give a summary of guidelines for choosing potential additives in the future. For example, is high optical activity good or bad? how can we increase the nematic range in temperature?
4. This study would be most useful if one or more of the mixtures show excellent characteristics as a nematic cell suitable for display or other optoelectronic applications. The present manuscript does not provide such data.
There are numerous grammar errors in the manuscript.
Author Response
Dear colleague,
First of all, we would like to express our thanks for the very careful and informative reviews. After getting the reviews we have used the template for the publication submission and made changes in accordance with journal requirements. As it is shown in the template we have moved the experimental part after introduction under number 2 (2. Materials and methods). We have added the part of new additive synthesis in “Materials and methods”.
- The authors have studied three new chiral substances as additives to a nematic liquid crystal. The difference in the optical activity and the chemical structure of additive molecules results in the appearance of the chiral nematic phase, in the change in both compatibility of mixture components and the temperature range of a liquid crystal phase. The paper is well-written, in general. The findings were supported by extensive measurements with a variety of instrumentation.
Response: Thank you.
- A survey of chiral additives that have been studied should be added. What is the logic of leading to the selection of these additives?
Response: We are currently running a general project on quantum dots having the chiral properties and their capability to induce chirality of nematic liquid crystals. That is why our choice is for chiral dopants carrying COOH group and capable acting as ligands ionically attached to QD surface. The first step after we have synthesized new systems we are checking their influence on a nematic liquid crystal. We did not yet publish the whole list of newly synthesized compounds and made our choice for three of them mentioned in our current manuscript.
We did not mention the above goal yet because this paper has its own goal: to study the effect of newly synthesized three chiral additives (CAs) having different molecular shape, size and optical activity on the optical, thermal and dielectric properties of a nematic (N) liquid crystal when mixed.
- The authors showed that the additives resulted in nematic mixtures with somewhat different characteristics. What have we learned from these studies? The authors should give a summary of guidelines for choosing potential additives in the future. For example, is high optical activity good or bad? how can we increase the nematic range in temperature?
Response:
Lines 386-396
The additive molecule having the small size (CA1) provides the maximal compatibility with LC matrix and demonstrates the increase in the temperature of the nematic-isotropic transition. Among a wide range of known chiral dopants that induce twisting of the nematic phase of a low molecular weight LC, the latter is very unusual and extremely rare. In most cases if the size of the additive molecules is bigger, than that of molecules of the matrix, N-I temperature falls down with the increase in the additive content. It corresponds to the other CA2 additive. However, its higher optical activity results in the selective reflection of light in the visible wavelength range indicating the stronger twisting of cholesteric pitch. High optical activity of the third additive (CA3) allows the helical twisting at a very low its content but the difference in the molecular structure provides the strong limitations of the compatibility.
We have mentioned the above paragraph in the Conclusion part of the new version of the current
Manuscript.
- This study would be most useful if one or more of the mixtures show excellent characteristics as a nematic cell suitable for display or other optoelectronic applications. The present manuscript does not provide such data.
Response: Your advice is generally right. However, as the main reason of our bigger project is to check the ability of chiral additives having different structures to provide chirality to QDs. What is done in that paper will be compared with the chiral quantum dots if any. Further we are checking the ability of new chiral QDs to be compatible with a nematic liquid crystal, to induce the QDs chirality transfer to the whole nematic phase, and to affect the time-on and time-off parameters suitable for display or other optoelectronic applications in future.

Reviewer 2 Report
The authors present the effect of three new chiral dopants on conventional thermotropic nematic liquid crystals. Depending on their molecular complexity they induce different degrees of chirality, increase or decrease the liquid crystal stability range, induce smectic ordering, and change surface anchoring from homeotropic to planar. The interesting variety of possible properties of these mixtures was unfortunately presented without enough explanations of observed details and providing an understanding of the presented phenomena. On top, the paper is written for specialists in POM of liquid crystalline textures. The paper needs to be substantially improved before I can recommend publication in Materials. Below I list my remarks and suggestions for improvements.
Line 29
A lot is going on in the field of chirality transfer, therefore some recent references should be included.
Lines 36-61 & Table 1
Add a comparison with well-known dopants.
Fig 3
It would be good to add experimental pitch values as functions of temperature and concentration. How their behavior relates to proposed ones. There is a lot of literature on dopant-induced chirality some are more than 40 years old like Chilaya’s 1981 paper.
Line 93
Structures in thin liquid crystalline layers are crucial for the presented study. Therefore, first, a clear explanation of the nematic anchoring direction and strength on confining surfaces is needed (Now it comes later and is not coherent!). The explanation of anchoring effects on structures should follow, etc. Here I mention just some relevant points. A homeotropic anchoring if it is strong prevents chirality-induced twisting in thin layers like in Fig.4a. If with doping homeotropic achoring turns to a planar one, probably the dopant modifies the surface anchoring. For selected reflection studies and pitch determination the best is degenerate planar anchoring. The use of oriented planar anchoring provided by rubbing is not OK as it does not allow optimal twisting. So the observed reflection does not relate to intrinsic pitch.
Lines 94-97 and 151
It should be clearly stated that there is a structural transition, not an N to N* transition. Dopant-induced N* is unwounded by homeotropic confinement for distances shorter than pitch.
Fig 4, 5, 7, 10, and related text
Displaying POM images requires explanations of presented textures and detailed features. The Materials journal is intended to address a broader audience not only LC texture specialists!
Also, information about crossed polarizers is missing on all the above-mentioned Figs.
Fig. 5
It would be good to comment on why the colors of the presented textures differ from the ones in Fig.4.
Lines 105-107
The statement ”… the increase in the CA2 content demonstrates the transition from high pitch chiral texture to the systems with the pitch selectively reflecting light in the visible spectral range.” needs explanation. Probably the high concentration of the dopant modifies the anchoring to a (degenerate?) planar one. Therefore a selective reflection is observed.
Fig. 6
Include pitch in the (b)!
Line 119
Add more details on the mentioned phase separation.
Lines 126-131
This discussion should appear earlier when texture studies start. It should include a comment on strength and degeneracy in the planar case.
Lines 138-140
A comment on the change of anchoring from planar to homeotropic with temperate should be added.
Fig. 9
Why the LMN curve differs from the one in Fig.8?
Line 161
The term oil stripes should be introduced probably at the beginning when a discussion of textures starts.
Line 142-172
The discussion of the planar sample with CA1 dopant is confusing. If the planar anchoring would be a degenerate planar type, the cholesteric would not be constrained and twist around the normal of the surface would adapt due to temperature and dopant concentration. If planar anchoring is unidirectional on both surfaces the cholesteric is constrained and would with changing temperature and concentration exhibit various twisted structures. This should be discussed!
Fig 11
Coordinate axes should be labeled.
Line 176
The confocal texture is not explained. Probably it would be better to use the term smectic texture.
Lines 222-239
For POM, selective reflection, and dielectric measurements anchoring on surfaces should be specified.
Lines 255-284
References are not uniformly presented. Only some papers have titles, somewhere the year is missing, some have starting and end pages, etc.
Mostly it is fine! There are some points where not the most appropriate words are used.
Author Response
Dear colleague,
First, we would like to express you our thanks for the very careful and informative review. Unfortunately we have not received your review before because of the journal problems to organize the list of reviews, which should contain four reviews instead of just two of them (3 and 4??) available for authors. We have done many changes in the text in accordance with advices of reviews available before. In general, after getting the reviews we have used the template for the publication submission and made changes in accordance with journal requirements. As it is shown in the template we have moved the experimental part after introduction under number 2 (2. Materials and methods). We have added the description of new additive synthesis in “Materials and methods”.
1.The authors present the effect of three new chiral dopants on conventional thermotropic nematic liquid crystals. Depending on their molecular complexity they induce different degrees of chirality, increase or decrease the liquid crystal stability range, induce smectic ordering, and change surface anchoring from homeotropic to planar. The interesting variety of possible properties of these mixtures was unfortunately presented without enough explanations of observed details and providing an understanding of the presented phenomena. On top, the paper is written for specialists in POM of liquid crystalline textures. The paper needs to be substantially improved before I can recommend publication in Materials. Below I list my remarks and suggestions for improvements.
Response: It is clear that the reviewer is a high-level specialist in liquid crystals and in display technologies. However, the main goal of our manuscript is not devoted to the comparison of newly synthesized additives with previously existing chiral additives for liquid crystals. We have synthesized these dopants with COOH groups in order to create chiral ligands capable of interacting with the surface of quantum dots for further incorporation into liquid crystals. That is why all three molecules contain COOH-groups. We understand that it is a long story, but we have to characterize these systems as chiral dopants before comparing them with chiral modified quantum dots inserted into a liquid crystal. If the reviewer insists, we will add this explanation to our introduction, but we hope that the reviewer understands the reason why this has not been done.
- Line 29
A lot is going on in the field of chirality transfer therefore some recent references should be included.
Response: We have added recent references on the chirality transfer in liquid crystals in the version 2 of our manuscript.
- Lines 36-61 & Table 1
Add a comparison with well-known dopants.
Response: As we have already mentioned above, the goal is to analyze the properties of new chiral dopants that we synthesized and to detect how the difference in their chemical structure will affect their ability to be dissolved by a certain nematic liquid crystal, to change the LC temperature range, and to twist the nematic structure.
That is why we think that there will be the excess of the information in our experimental table if we add info on the different systems.
- Fig 3
It would be good to add experimental pitch values as functions of temperature and concentration. How their behavior relates to proposed ones. There is a lot of literature on dopant-induced chirality some are more than 40 years old like Chilaya’s 1981 paper.
Response: You are right. We know this paper, as well as of many others published during the last 40 years. However, as we have explained in the first response, we synthesize our systems to use them further as chiral ligands. However, we had to check their ability to create chiral nematic phase. That is why the measurements of the pitch values are not necessary right now. However when we finalize the synthesis of more new chiral additives with functional groups, we will give the total description with pitch values as well.
- Line 93
Structures in thin liquid crystalline layers are crucial for the presented study. Therefore, first, a clear explanation of the nematic anchoring direction and strength on confining surfaces is needed (Now it comes later and is not coherent!). The explanation of anchoring effects on structures should follow, etc. Here I mention just some relevant points. A homeotropic anchoring if it is strong prevents chirality-induced twisting in thin layers like in Fig.4a. If with doping homeotropic achoring turns to a planar one, probably the dopant modifies the surface anchoring. For selected reflection studies and pitch determination the best is degenerate planar anchoring. The use of oriented planar anchoring provided by rubbing is not OK as it does not allow optimal twisting. So the observed reflection does not relate to intrinsic pitch.
Response: The interaction of liquid crystals with the surfaces in cells undoubtedly influences the optical textures in thin layers. In our case, it is evident that the twisting of the nematic LCs occurs with difference in the textures being observed when the LCs are brought to the same temperature by being heated or being cooled. Taking into account that in a majority of POM experiments we did not treat the cell surfaces, the influence of the chiral additive on a nematic matrix being cooled from an isotropic liquid state should be much stronger than when heated from room temperature. We have discussed it in more details in the second version of our manuscript (discussion about Figure 4).
We have not measured the intrinsic pitch but need just the information on the twisting of a nematic structure by chiral additives having different chemical structure.
- Lines 94-97 and 151
It should be clearly stated that there is a structural transition, not an N to N* transition. Dopant-induced N* is unwounded by homeotropic confinement for distances shorter than pitch.
Response: It is right: N to N* is a structure change induced by a chiral additive. But the DCS fixes N-I transition for mixtures as N* - I.
- Fig 4, 5, 7, 10, and related text
Displaying POM images requires explanations of presented textures and detailed features. The Materials journal is intended to address a broader audience not only LC texture specialists!
Also, information about crossed polarizers is missing on all the above-mentioned Figs.
Response: To satisfy the reviewer requirement we have made changes and additions in the second version of manuscript:
Lines 218 – 254.
The analysis of the optical textures obtained in crossed polarizers in cells without surface treatment shows that the layer of LMN has a homeotropic texture with the long molecular axis being oriented preferably normal to the sample plane (Figure 4a). The addition of CA1 in amounts creating mixtures with a concentration below 10 wt.% , does not change the homeotropic texture, even if the clearing temperature TNI is higher (Figure 3a, curve 1). However, the cooling of that sample to below the TNI of the mixture (70oC) results in the appearance of an oil stripes texture. (Figure 4b). At a CA1 content below 25 wt.% the solubility of the initially crystalline CA1 is sufficiently high and is preserved at room temperature, indicating the weak influence of the chiral dopant on the nematic matrix and the preservation of the homeotropic texture. Roughly speaking, the anchoring energy of the nematic is higher than the twisting power of CA1. Nevertheless all samples with CA1 content below 25 wt.% undergo a violation of the homeotropic orientation under heating up to the vicinity of TNI of LMN matrix. The higher the content of CA1, the lower the temperature of the texture change occuring. As for the mixture with 25 wt.% of CA1, the homeotropic texture at room temperature is distorted and small light structures appear(Figure S6), which are growing into chiral texture (Figure 4e). Generally, at CA1 concentrations between 15 and 25 wt.% cooling below the TNI of the mixtures one observes polydomain textures being formed by the chiral nematic structures and the chiral twisting being stronger than that under heating (Figure S7). Contrary to the LMN-CA1 mixtures TNI transition of mixtures with CA2 decreases (Figure 3b, curve 2) with the increase in CA2 content. However those mixtures starting with 2 wt.% of CA2 have the chiral twisted texture at room temperature (Figure 5). The presence of CA2 seems to change the anchoring of matrix molecules, which transforms the homeotropic texture. This effect, in combination with the higher optical activity of CA2 chiral molecules (Table 1), allows the selective reflection to appear in the visible spectral range.
As an example, the Figure 6a shows the curves related to the selective reflection of light of the LMN with 30 wt.% of CA2 [18]. When the content of the additive reaches 20 wt.% the selective reflection happens at 850-950 nm (Figure 6b). At the 30 wt. % of CA2 the selective reflection is observed at 500-600 nm, whereas at 40 wt.% it shifts down to 400-500 nm.
- Fig. 5
It would be good to comment on why the colors of the presented textures differ from the ones in Fig.4.
Response: small changes in the molecules orientation in an anisotropic medium leads to an essential phase shift. This causes changes in the colors observed in crossed polarizers. Due to Figures 4 and 5 being about different systems it is difficult to expect identical colors.
- Lines 105-107
The statement ”… the increase in the CA2 content demonstrates the transition from high pitch chiral texture to the systems with the pitch selectively reflecting light in the visible spectral range.” needs explanation. Probably the high concentration of the dopant modifies the anchoring to a (degenerate?) planar one. Therefore a selective reflection is observed.
Response: Thank you. The reviewer’s idea seems probable. At a content of CA2 additive above 5wt. %, a more stable planar texture appears and the selective reflection may be observed. The formation of a planar texture may result from the dopant’s modification of the anchoring of the initial matrix with the cell surface. That is why the twisting effect becomes stronger.
- Fig. 6
Include pitch in the (b)!
Response: As we have already mentioned above, the major goal of the manuscript is not directed at the full description of optical properties.
- Line 119. Add more details on the mentioned phase separation.
Response: We have changed the text in accordance with the advice:
Lines 262 – 267. Chiral texture exists for all studied mixture concentrations (Figure 7), but at 6 wt. % of CA3 and higher one may observe the appearance of dark areas- amorphous areas of CA3 liquid. This means that the compatibility of the components has become poor. The maximum saturation of CA3 by a nematic matrix seems to be about 5 wt.%. The concentration range where CA3 shows full compatibility with LMN is quite narrow.
- Lines 126-131
This discussion should appear earlier when texture studies start. It should include a comment on strength and degeneracy in the planar case.
Response: we have already included this discussion in p.6 Lines 218 – 236.
12.Lines 138-140
A comment on the change of anchoring from planar to homeotropic with temperature should be added.
Response : We have changed the text . See below:
Line 283 -293. As is observed from the optical textures, mixtures with CA2 may form initial planar orientation similar to those with CA3. This is confirmed with the curve e¢/e¢is (T) of 5 wt.% mixture with CA2 (Figure 8b). At a higher CA2 content (about 20 wt.%) the dielectric permittivity, while being heated, behaves in a similar way. However during the cooling of the sample from the isotropic phase the value of the e¢/e¢is begins to increase below TNI. Notably, the LC optical texture practically does not change when the mixture is heated or cooled (Figure S8). It seems that this effect may be related to the orienting influence of the field signal level on the LC composition with a positive dielectric anisotropy. Along with an increase in CA2 content, which induces stronger chiral twisting, the anchoring effect becomes much weaker. This results in a partial molecular reorientation along the field direction during the cooling from the isotropic phase.
- Fig. 9
Why the LMN curve differs from the one in Fig.8?
Response: With this Figure 8 we have intended to pay attention on the correlation between the change in optical textures and the appearance of peaks on the temperature curve of the normal component of the dielectric permittivity. That is why we renormalize the axis for the better visualization of the effect observed.
- Line 161
The term oil stripes should be introduced probably at the beginning when a discussion of textures starts.
Response: It is done for the discussion of Fig.4 Lines 218 – 224.
- Line 142-172
The discussion of the planar sample with CA1 dopant is confusing. If the planar anchoring would be a degenerate planar type, the cholesteric would not be constrained and twist around the normal of the surface would adapt due to temperature and dopant concentration. If planar anchoring is unidirectional on both surfaces the cholesteric is constrained and would with changing temperature and concentration exhibit various twisted structures. This should be discussed!
Response: We have this discussion in the text:
Lines 305 – 361. The conditions of the experiment are as follows: the optical textures and the dielectric measurements were analyzed simultaneously in cells with unidirectionally treated surfaces.
The analysis of the optical textures as a function of the dopant content allows us to hypothesize that at concentrations below 25 wt.% the planar unidirectional anchoring is preserved on both surfaces. The mixtures exhibit very similar twisted structures with changing temperature and concentration. At higher concentrations like 25 wt.% and above the set of various optical textures becomes broader.
It may be likely the planar anchoring is a degenerate one.
The initial homogeneous texture of the LMN matrix (Figure 10a) does not really change when one inserts 10 wt.% of CA1 (Figure 10b). When approaching TNI of this mixture the chiral texture appears, which exists in a very narrow temperature range (Figure 10c). The increase in CA1 content up to 20 wt.% also does not change the initial planar orientation (Figure 10 d). However, when approaching TNI of the initial nematic matrix (64oC) by heating, the typical twisted texture appears with the axis located in-plane of the cell (Figure 10e). One may suppose that at that moment, the interaction between the surface and mesogenic molecules is becoming weaker. At the same time, a small peak in the e¢/e¢is(T) curve (Figure 9) appears. With a further increase in CA1 content up to 25 wt.% the peak becomes more pronounced and a second peak shows up in the vicinity of 72oC. The corresponding optical textures are shown in Figure 10 f-j. The analysis of the textures lets us conclude about the transformation of N structure of the mixture to a more ordered one (Figure 10f). Above 64oC the ordered texture is transformed to a chiral one, which is also changing with the temperature (Figure 10g,h). Note that in all cases this system forms the planar orientation with the characteristic texture known as oil stripes (Figure 10h). At 72oC the peak on the e¢/e¢is(T) curve indicates the next transformation of a chiral texture, accompanied by the relocation of the twisting axis in to the plane of the cell (Figure 10i,j). The cause of the change in the location of the axis from the normal to a parallel cell surface is not yet clear. One of the possible explanations may be the stronger twisting effect from heating (Figure 10i). When approaching the TNI of the 25 wt.% composition the texture changes again to the planar one with characteristic oil stripes (Figure 10j).
As for the mixture with 30 wt.% of CA1, its curve has only one peak (Figure 9) at about 70oC. It corresponds to the transformation of the smectic-like fan texture (Figure 10k), existing in a broad temperature range, in to a different one, which resembles the blue phase texture (Figure 10l). The latter endures the transition to a planar textures with oil stripes, preceding the transition to an isotropic phase at 85oC (Figure 10m).
The analysis of the optical textures as a function of the dopant content allows us to suppose that at concentrations below 25 wt.% the planar unidirectional anchoring is preserved on both surfaces, and the mixtures exhibit very similar twisted structures when undergoing a change of temperature and concentration. At higher concentrations, like 25 wt.% and above, the set of various optical textures becomes broader. It may be likely that the planar anchoring is a degenerate one because of the increase in CA1 dopant content.
The texture transitions discussed above are in good agreement with DSC data (Figure 11). There are two transitions observed: during heating the first transition corresponds to 72oC with a transition enthalpy of 4 Jg-1, the second one proceeds at 84.6oC with a corresponding enthalpy of 5.5 Jg-1 (curve 1). These transitions are fully reversible, as shown in Figure 11, curve 2.
Thus, using POM and DSC results we observe the transition of the 30 wt.% system with a fan texture to the chiral phase, which in its turn is transformed into the isotropic phase. To analyze the observed transition we have used small angle X-ray scattering. Curves, obtained at various temperatures, are given in Figure 12. One can observe an X-ray Bragg reflection with the corresponding d-spacing being equal to 2.31 nm at room temperature. This d-spacing is roughly the size of a pair of CA1 molecules connected by a hydrogen bond, supporting evidence of a layered mesophase presence.
- Fig 11
Coordinate axes should be labeled.
Response: Thank you. DONE
- Line 176 he confocal texture is not explained. Probably it would be better to use the term smectic texture.
Response: You are right. We have followed your advice.
- Lines 222-239
For POM, selective reflection, and dielectric measurements anchoring on surfaces should be specified.
Response: The problem of the interaction of LC medium with the surface is very important. It is particularly true from the viewpoint of the new systems usage in display technologies. As we have already mentioned above the main goal of this manuscript is to demonstrate the principle ability of COOH-containing chiral systems to induce the chiral twisting of LC phase in different concentration ranges, their compatibility with a nematic matrix and the influence on the N-I transition temperature.More details of the experiment are given in the part 2 of the modified version.
- Lines 255-284
References are not uniformly presented.
Response: Thank you. DONE

Reviewer 3 Report
In this paper, the authors reported the functional effect of three chiral molecules as additives on the formation of chiral textures in nematic liquid crystal. The optical images clearly showed such effect. I think the main results are clearly presented and properly discussed. But after reading through the text, I still not quit understand the relation between the optical activity of the additives and the chiral textures in the liquid crystals. I think this paper has to be extensively revised before a further consideration for publication. My suggestions are given below.
(1) Either optical rotatory dispersion spectra or circular dichroism spectra for the three additives should be provided. Or, the key word in the title “optical activity” would be better to remove. When it is mentioned with a specific value, it must be noted at which wavelength of light this value was measured.
(2) Any acronym should be defined before its first use, e.g., DSC, TNF, LMN.
(3) The bibliography should be carefully edited.
(4) I suggest add specific discussions on related and/or potential applications in the Abstract and Introduction part.
Please avoid using non-English letters (e.g., on Line 65).
Author Response
Dear colleague,
First of all, we would like to express our thanks to our reviewers for the very careful and informative reviews. After getting the reviews we have used the template for the publication submission and made changes in accordance with journal requirements. As it is shown in the template we have moved the experimental part after introduction under number 2 (2. Materials and methods). We have added the part of the additive synthesis in the Materials and methods part.
- Either optical rotatory dispersion spectra or circular dichroism spectra for the three additives should be provided. Or, the key word in the title “optical activity” would be better to remove. When it is mentioned with a specific value, it must be noted at which wavelength of light this value was measured.
Response: Thank you for the advice. We have followed your advice and removed “optical activity” from the title. In addition we have mentioned the methods of the measurement and added necessary notation in the Table.
- Any acronym should be defined before its first use, e.g., DSC, TNF, LMN.
Response: DONE
- The bibliography should be carefully edited.
Response: DONE
- I suggest add specific discussions on related and/or potential applications in the Abstract and Introduction part.
Response: We have used the general words about application of such systems in the introduction. It is an important point. However in order to finalize the project we are carrying out we have synthesized a bigger set of additives, which will help to make a certain application choice.
Reviewer 4 Report
Please see the attached file.

Author Response
Dear colleague,
first of all, we would like to express our thanks you for the very careful and informative reviews. After getting the reviews we have used the template for the publication submission and made changes in accordance with journal requirements. As it is shown in the template we have moved the experimental part after introduction under number 2 (2. Materials and methods). We have added the part of the new additive synthesis in the Materials and methods part. Due to the changes made the total number of pages in the version two is larger now than that in the initial version. That is why our answers correspond to the pages numbered in the version two. We have slightly changed the title in accordance with the reviewer’s 1 advice.
- First of all, the authors must revise the manuscript to conform to the format of “materials”. (p.1, lines 4–5) Authors’ full names should be used. The authors’ names should appear in the order of “first name”, “middle name”, and “family name”. (p.11, lines 232–239, p.11, line 255–p.12, line 284) Font size should be changed.
Response: We have changed the font in accordance with the template and used template to present the paper. To satisfy the format of “Materials” we have done the changes suggested by reviewer. For example:
- 1 Alexey S. Merekalov1, Oleg N. Karpov1, Georgiy A. Shandryuk1, Olga A. Otmakhova1, Artem V. Bakirov2, Vladimir S. Bezborodov3 and Raisa V. Talroze1,*
- 2-4 In addition we have made the change in the structure of paper and placed the experimental part “Materials and Methods” right after the Introduction.
- (p.1, lines 33–35) Why did the authors select those three chiral dopants in a lot of chiral compounds. Please briefly explain the motivation and their notable features.
Response: First of all the new chiral additives are quite NEW. To clarify the situation we have added new sentences marked with red in the Introduction part.
p.1-2 The major goal of the current research work is to study the effect of newly synthesized three chiral additives (CAs) having different molecular shape and optical activity on the optical, thermal and dielectric properties of a nematic (N) liquid crystal when mixed. Optical activity is the property of a compound being able to rotate the plane of polarization of plane-polarized light, and a compound with such activity is labelled as optical active. We have chosen a new set of chiral substances, namely, R(+)2(methyl-butoxy)benzoicacid (CA1), R(+)-2-(4"-hexyl-2'-chlorine-[1,1':4',1"-ter- phenyl- 4-iloxy)propanoic acid (CA2) and R(+)6,6'-([1,1'-binaphthalene]-2,2'-diylbis(oxy)) dihexanoic acid (CA3) (Figure 1). As we show in this paper these systems are optically active and we intend to figure out their ability to induce the chiral structure of the low molecular nematic matrix (LMN), which is the mixture of several derivatives of 4-cyanobiphenyl. This matrix has a broad nematic temperature range, and it is widely used in our researches. We have presented the whole description of LMN composition in [18]. The major difference in their molecular structure dictates the necessity of the analysis of both the principal compatibility of components on the phase behavior and other physical chemical properties of the resultant mixtures. One expects that in the first turn it will affect the optical properties of new LC mixtures.
3.(p.1, lines 36–38) The absolute configurations of chiral dopants must be shown. Please show the basic characterization data of 1H NMR, 13C NMR, MS, and elemental analysis apart from FT-IR spectra
Response: The analytical data are placed in the Supporting Information file.
- 1 (abstract and introduction.). The absolute configurations of chiral dopants is also shown.
p.2 and 3. We have added the detailed description of synthesis of new additives and presented it in “Materials and methods” part.
Supporting file. We have used different spectra analysis, and some examples of FTIR, 1H NMR, 13C NMR, MS are summarized in Figures S1-S4.
- (p.1, line 41) How did the authors obtain the LMN? Please describe. If the authors are not possible to show the composition of LMN, the product name or code of the liquid crystal mixture should be listed.
Response:p.2 The abbreviation of the nematic used is LMN. The latter is the mixture of several derivatives of 4-cyanobiphenyl. This matrix has a broad nematic temperature range, and we have already used and described it in [18]
- (p.2, line 49) What are the TNF solutions? Is the solvent tetrahydrofuran (THF)?
Response: p.4 Yes, you are right.
- (p.2, Figure 2) The vertical and horizontal axes should be labeled “heat flow” and “temperature”, respectively. Furthermore, the direction of exothermic flow should be shown.
Response: p.5 Done
- (p.3, Table 1) The authors should show the phase transition behavior of LMN without chiral dopant in Table 1.
Response: p.5 The phase transition behavior of LMN without chiral dopants is shown in Table 1.
In addition, the experimental evidence of the phase transition behaviors of CA1 and CA2 are required. I cannot verify the assignment of their crystalline phases without any POM images and XRD profiles. The unit notation rule should be unified. I recommend that the unit of “J/g”, “deg 1/(M dm)” and “deg mL/(g dm)” must be changed to “J mol 1 ”, “deg M 1 m 1 ” and “deg mL g 1 dm–1 ”, respectively. The notation of phase transition sequence is desirable to imitate the style of Table 1 in J. Phys. Chem. B2015, 119, 11935–11952.
Response: p.5 Table 1. We have also changed the units and the notation of the transition sequence in accordance with the reviewer’s comment
- (p.3, line 65) Please check not to mix Russian language.
Response: Sorry, we have not found it.
- (p.3, Figure 3) I think that the content of chiral dopants should be displayed as molar content (mol%), because the comparison by the molar content equivalent to a comparison by the number of molecules.
Response: p. 6 This is a very good advice. However due to the nematic matrix (LMN) presented is a mixture of several compounds our decision is to leave it as is.
- (p.3, lines 78–79) The authors should discuss that what kind of interactions were effective in this study. What is the origin of “additional interaction”?
Response: We have additionally analyzed the set of papers published and made a kind of change in our text. Please see below the text placed in in pp. 5-6 (new version).:
- 5.-6. The increase in TNI observed in mixtures with CA1 indicate the possibility of the increase in order caused by the formation of molecularly rigid and elongated dimers of CA1 in the nematic LC, as confirmed with infrared spectra. CA1 contributes in the increase of the transition enthalpy (from 3.4 up to 6.5 Jg-1) (Figure 3b, curve 1). At the same time one may see in LMN mixtures with CA2 just as continues drop of TNI without any visible destroying the LC structure, although the corresponding enthalpy decreases from 3.4 down to 2.2 Jg-1 (Figure 3b, curve 2). It is a clear indication that the CA2 additive destroys LMN possibly due to a big size of its molecules in comparison with the nematic matrix molecules. These data are in a good consistency with a quantitative correlation established in [19]. They show the doping of the liquid crystal with molecularly flexible acids to cause the lowering of TNI. The increase in TNI happens if the rigid carboxylic acids form dimers comparable in size with the nematic matrix molecules.
- (p.5, Figure 6a) The label of vertical axis should be “Transmittance”.
- 8 Response: DONE
- (p.5, Figure 6) Please attach a scale line.
Response: DONE
- (p.6, line 124) What is referred “Table A3” ? I could not find in the manuscript and ESI.
Response: We apologize for this untidy error. We change the error with the following sentence:p.8 “The concentration range, where CA3 shows the full compatibility with LMN is quite narrow. Such a poor compatibility results from the very different molecular structure of components.
- (p.11, line 247) The value of pressure should be described as 6.7 × 10-3 mmHg or 0.89 Pa.
Response: p.4 DONE
- (Introduction and Conclusion) Various chiral dopants have been developed and been evaluated the influence upon the induction of chiral liquid-crystalline phase. The authors must compare this study to previous studies. Then, the authors should argue the novelty and originality of this study in the manuscript. I think that the description in the present manuscript is inadequate.
Response: Thank you for your advice, We have added some discussions and references in the new version of the manuscript.
- Helical twisting power (HTP) is an important performance indicator for chiral dopants. Therefore, the study on HTP may provide new insight and may show the superiority of this study. Summary: The concept described in this article will attracts broad research interests for the expert of liquid crystals. While the authors have been evaluated three-types of chiral dopants, the discussion described in present manuscript seems to be insufficient.
Response: You are right. We are going to measure HTP in the next step of our project when the whole set of quite new chiral additives synthesized will be analyzed in full.
Round 2
Reviewer 1 Report
The response of authors are acceptable. However, I would suggest incorporation of responses to my comment, point (4), into the manuscript, perhaps in the introduction.
Author Response
Dear Reviewer,
Thank you for your advice. Please see below the sentence we have included in the Introduction.
Looking forward we are considering these systems to be effective in the case consider the possibility of using these mixtures as materials for electro-optic displays if additives provide a strong twisting of the nematic matrix and preserve the necessary orientation control in the cell. Moreover, we consider new additives as possible ligands, transferring the chirality to quantum dots, ensure their compatibility with nematic liquid crystals and provide the reasonable effect on switching times of optoelectronic effects.

Reviewer 3 Report
I am glad to see the authors accepted my suggestions and revised their paper accordingly. I think this paper has been improved a lot and is ready for publication. No re-review would be needed from me.
Author Response
Dear Reviewer,
Thank you so much for your kind response.
Yours sincerely, authors
Reviewer 4 Report
Please see the attached file.

Author Response
Dear colleague,
First of all, we would like to express our thanks to the reviewer for the very careful and informative review. Unfortunately, we have not had this comment before because some problems online , and after receiving it we spent extra time on purification of our samples and additional analysis and testing.
Comments on the manuscript:
- (p.2, line 66– p.4, line 110) Compound names should be unified with IUPAC names. Hyphens should be inserted where appropriate positions.
Response: You are right and made changes:
4-[(2S)-(+)-2-methylbutoxy]benzoic acid
(2R)-(+)-2-[4-[2-chloro-4-(4-hexylphenyl)phenyl]phenoxy]propanoic acid
6-[1-[2-(5-carboxypentoxy)naphthalen-1-yl]naphthalen-2-yl]oxyhexanoic acid
- (ESI, Figures S2–S4) In these 1H NMR spectra, some signals originated from impurities are found. I think that these samples do not satisfy the purity criteria indicated in the submission rules. The purity is very important factor for liquid-crystalline (LC) phase transition behaviors. Therefore, I strongly recommend the authors for additional purification and vacuum drying. The subsequent remeasurement of NMR spectra and reexamination of LC properties should be required. In addition, I would like to request the authors to carry out the elemental analysis.
Response: We have carried out vacuum drying of our samples and the elemental analysis (see an addition in the part “Materials and methods”), obtained new 1H NMR spectra and checked the main characteristics of LC mixtures. As one can see from Figures S2a, S3 and S4/ The data obtained satisfy the purity criteria.
CA1 - Calculated (%): C 69.21; H 7.74; O 23.05; Found (%): C 68.95; H 7.92; O 23.13.
CA2 - Calculated (%): C 74.21; H 6.69; Cl 8.11; O 10.98; Found (%): C 74.35; H 6.76; O 11.10.
CA3 - Calculated (%): C 74.69; H 6.66; O 18.65; Found (%): C 74.60; H 6.78; O 18.62.

Round 3
Reviewer 4 Report
Please see the attached file.

Author Response
This original article describes the influence of chiral dopants upon the mesomorphic properties of a host nematic liquid crystal. The concept described in this article will be helpful to the development of chiral dopants. Although the revised manuscript still includes some notational issues such as the inappropriate font style, the content itself is deemed appropriate. Therefore, I recommend that the manuscript should be acceptable after minor revision.
Comments on the manuscript:
- (p.2, lines 70, 74) The letters of “R” and “S”, which indicate absolute configuration, should be written in italic.
- (p.3, line 88) The configuration of axis chirality for CA3 must be determined.
- (p.5, line 163; p.6, lines 191, 192, 193, 200, 205, 222, 223) The letters of physical quantity should be written in italic. “Tg” and “TNI” must be “Tg” and “TNI”, respectively.
- (p.5, line 181) Please remove Russian font of “и” between “2870-2960 сm-1” and “1460 сm-1” and please translate to English.
- (p.5, Table 1) The description of unit should be plain font. It should not be written in italic. The notation rule of unit should be unified. The notation of “Jg-1” and “J*g-1” should be corrected to “J g-1”. The other unit description of molar optcal activity and specific optical activity should be also changed in the same way.
- (p.6, lines 198, 201) Please insert the space between “J” and “g-1”.
Summary:
The concept described in this article will attracts broad research interests for the expert of liquid crystals. While some notation errors are still found in the revised manuscript, I think that the content was improved and satisfy the quality for publication. Thus, I recommend that the manuscript should be acceptable after minor revision.
Response:
The configuration of CA3 molecules S-(+)-6-[1-[2-(5-carboxypentoxy)naphthalen-1-yl]naphthalen-2-yl]oxyhexanoic acid is determined.
We apologize for some notational issues, and made all corrections advised.
Thank you!
